# Performance Analysis of a Solar Heating Ammonia Decomposition Membrane Reactor under Co-Current Sweep

**DOI:** 10.3390/membranes12100972

**Published:** 2022-10-03

**Authors:** Tianchao Xie, Shaojun Xia, Jialuo Huang, Chao Wang, Qinglong Jin

**Affiliations:** College of Power Engineering, Naval University of Engineering, Wuhan 430033, China

**Keywords:** ammonia decomposition, solar energy, energy storage, hydrogen production, membrane reactor

## Abstract

Ammonia is an excellent medium for solar thermal chemical energy storage and can also use excess heat to produce hydrogen without carbon emission. To deepen the study of ammonia decomposition in these two fields, finite-time thermodynamics is used to model a solar-heating, co-current sweeping ammonia decomposition membrane reactor. According to the needs of energy storage systems and solar hydrogen production, five performance indicators are put forward, including the heat absorption rate (HAR), ammonia conversion rate (ACR), hydrogen production rate (HPR), entropy generation rate (EGR) and energy conversion rate (ECR). The effects of the light intensity, ammonia flow rate, nitrogen flow rate and palladium membrane radius on system performances are further analyzed. The results show that the influences of the palladium membrane radius and nitrogen flow rate on reactor performances are very slight. When the light intensity is increased from 500 W/m^2^ to 800 W/m^2^, the ACR, EGR, HAR and HPR increase obviously, but the ECR decreases by 14.2%. When the ammonia flow rate is increased by 100%, the ECR, EGR and HPR increase by more than 70%, the HAR increases by 15.6% and the ACR decreases by 12.9%. At the same time, the ammonia flow rate needs to be adjusted with the light intensity. The results can provide some guiding significance for the engineering application of ammonia solar energy storage systems and solar hydrogen production.

## 1. Introduction

The ammonia decomposition reaction has a significant endothermic effect. The active temperature of the reaction is about 700 K, which is very suitable for a trough solar collector. Therefore, ammonia is an excellent medium for solar thermal chemical energy storage. At the same time, ammonia has great potential for producing hydrogen without any carbon emissions. So, using ammonia for solar energy storage can produce and sell hydrogen in the case of excess heat, realizing solar hydrogen production.

As for the research in the energy storage field, in 1992, Lovegrove et al. [1,2,3,4,5] of the Australian National University (ANU) solar group conducted experiments with an electrically heated ammonia reactor. He showed that the ammonia reactor could absorb up to 2 kW of heat. After that, Luzzi et al. [6,7] analyzed the performance of the ammonia decomposition reactor heated by a butterfly solar concentrator. He found that the peak endothermic power can reach 2.5 kW, and the solar energy absorption rate can reach 58%. In 1999, the ANU solar group built the first solar thermal storage system with ammonia as the circulating medium using a 20 m² butterfly collector, achieving a continuous output of 1 kW of power for 24 h [3,7,8,9,10].

The use of membrane reactors in the ammonia decomposition process can improve system performance. The existing research on ammonia decomposition membrane reactors mainly focuses on its influence on thermodynamic equilibrium. Wang et al. [11] applied ammonia decomposition membrane reactors to reduce the temperature of complete ammonia decomposition by about 200 °C under commercial Ni-based catalysts. Rizzuto et al. [12] conducted palladium membrane reactor experiments using the Ru catalyst and found that the ammonia conversion rate was the most stable when the temperature was 450 °C. The ammonia conversion rate was over 99.4% under all experimental conditions. Itoh et al. [13,14] found that the ammonia conversion of the membrane reactor was 15% higher than that of the conventional reactor at 723 K through theoretical modeling and experiments, and the hydrogen recovery was 60%. Cechetto et al. [15] achieved complete ammonia decomposition at 425 °C using a ruthenium-based catalyst and a palladium membrane and recovered more than 86% of hydrogen.

There is more engineering significance to analyzing and optimizing the ammonia decomposition membrane reactor. Abashar et al. [16,17] found an optimized ammonia feed point distribution of the membrane reactor using modeling, which can increase the reaction rate by 17.45% and reduce the reactor length by 75%. Wang et al. [18] modeled and studied one thermostatic ammonia decomposition membrane reactor heated by solar and found that the energy conversion rate can reach 40.08%. The energy conversion rate is defined as the ratio of incremental chemical energy to solar energy input. Xie et al. [19,20] established the solar endothermic system model for ammonia decomposition based on membrane reactors. The effects of factors such as the reactor radius, ammonia flow rate, light intensity and permeation zone pressure on system performance were analyzed.

In the existing research on the solar-heated ammonia decomposition membrane reactor, Wang et al. [18] used a thermostatic reactor to decompose ammonia and then converted the energy consumption into solar energy. This is not enough to meet the boundary conditions of practical engineering. The membrane reactor model of Xie et al. [19,20] utilizes the vacuum insulation layer outside the reactor for hydrogen permeation. This will result in a higher heat loss and thus affect system performance. To meet the demand of the engineer boundary conditions, reduce the heat loss and analyze the effect of reactor parameters on performance, this paper use finite-time thermodynamics [21,22,23,24] to establish an ammonia decomposition membrane reactor model with the outermost vacuum insulation layer, the middle annular reaction zone and the innermost layer a nitrogen-sweeping permeation zone. The effects of the ammonia flow rate, nitrogen flow rate, light intensity and radius of the permeation zone on the heat absorption rate (HAR), ammonia conversion rate (ACR), hydrogen production rate (HPR), entropy generation rate (EGR) and energy conversion rate (ECR) of the reactor are analyzed.

## 2. Physical Model

Figure 1 is the model schematic of a solar heating ammonia decomposition membrane reactor. Sunlight is directed onto the LS-3 concentrating collector [19] and then converges on the reactor, heating and raising the gas temperature in the reaction zone to drive the ammonia decomposition reaction.

### 2.1. Solar Collector Model

The collector converges sunlight to the outer wall of the membrane reactor to provide energy for the ammonia decomposition reaction. Table 1 shows the basic geometric and optical parameters of the LS-3 type solar collector.

According to the surface light intensity on the ground, the solar power density absorbed by the reactor can be converted as [19,20]:(1)$$P_{\mathrm{tube}}=P_{\mathrm{sun}}K_{s}\eta_{s}\delta_{s}\varepsilon_{s}$$
where *P*_sun_ is the average light intensity on the ground, in W/m^2^; *P*_tube_ is the power density of absorbed solar energy, in W/m.

### 2.2. Reaction Kinetic Model

The reaction zone is filled with a commercially available Ru/Al_2_O_3_ catalyst, and the rate equation for the ammonia decomposition obeys the Temkin–Pyzhev mechanism [25,26,27]:(2)$$R_{A}=K_{\mathrm{app}}\left[{\left(\frac{p_{A}{}^{2}}{p_{H}{}^{3}}\right)}^{\beta }-\frac{p_{N}}{K_{\mathrm{eq}}{}^{2}}{\left(\frac{p_{H}{}^{3}}{p_{A}{}^{2}}\right)}^{1-\beta }\right]$$
where *p_i_* is the partial pressure of each component in the reaction zone, in bar, and the different subscript *i* means a different component (A is ammonia, N is nitrogen and H is hydrogen); *K*_eq_ is the equilibrium constant; *K*_app_ is the speed constant of the ammonia decomposition reaction; *R_A_* is the reaction rate of ammonia decomposition; *β* is a constant which is related to the catalyst surface state.

*K*_app_ in Equation (2) is calculated by:(3)$$K_{\mathrm{app}}=k_{0}\exp(\frac{-E_{\mathrm{app}}}{R_{g}T})$$
where *k*_0_ is the pre-exponential factor, and *E*_app_ is the activation energy of the ammonia decomposition reaction of the catalyst.

In the study by Prasad et al. [27], they prepared an Ru/Al_2_O_3_ catalyst (with 4% Ru). To determine the reaction kinetic parameters, they used a flow controller to control the flow rates of ammonia, hydrogen, nitrogen and helium input into the reactor, measured the temperature distribution of the reactor using thermocouples and then used the experimentally measured data to refine the model parameters. Finally, they obtained the parameter values in Equations (2) and (3): *β* = 0.27, *k*_0_ = 6 × 10^8^ s^−1^ and *E_app_* = 117 kJ/mol.

### 2.3. Membrane Reactor Model

The cutaway view of the ammonia decomposition membrane reactor is shown in Figure 2. The reactor is divided into three areas by a three-layer structure. The outermost layer is the glass layer, which allows light to pass through; the middle layer is the outer wall of the reactor, which is coated with a special coating to improve the absorption rate of solar radiation and reduce the emissivity; the innermost layer is a palladium membrane with support structures, allowing hydrogen to pass through from the reaction zone to the permeation zone. The outermost annular area is a vacuum area, which can effectively weaken the convective heat loss of the reactor outer wall; the middle annular area is a reaction area, with the Ru/Al_2_O_3_ catalyst added, and the ammonia decomposition reaction is also carried out here; the innermost circular area is the permeation zone, where the hydrogen permeating from the reaction zone is swept by nitrogen in the co-current mode.

The basic parameters of the membrane reactor model are shown in Table 2.

The reactor model can be described by mass conservation, momentum conservation and energy conservation.

#### 2.3.1. Mass Conservation Equation

The change in the hydrogen flow rate can represent the mass conservation equation in the reaction zone well:(4)$$\frac{dF_{H}}{dz}=1.5S_{\mathrm{RA}}(1-\varepsilon_{p})R_{A}-J_{H}$$
where *F*_H_ is the hydrogen flow rate in the reaction area, in mol/s; *S*_RA_ is the area of the reaction zone; *J*_H_ is the permeability of hydrogen.

The hydrogen in the permeation zone comes entirely from permeation, so its flow rate change is:(5)$$\frac{dF_{H,P}}{dz}=J_{H}$$

The hydrogen permeability of the palladium membrane is [18,28]:(6)$$J_{H}=\frac{k(p_{H}^{n}-p_{H,P}^{n})}{dM}$$
(7)$$k=3.21\times 10^{-8}\exp(-\frac{13,140}{R_{g}T_{M}})$$
where *p*_H_ and *p*_H,P_ are the hydrogen partial pressure in the reaction zone and the permeation zone, respectively, in Pa; *T*_M_ is the temperature of the membrane, which is calculated by the temperature of the reaction zone and the permeation zone and the thermal resistance on both sides of the membrane; *n* takes 0.62 [18,28].

#### 2.3.2. Momentum Conservation Equation

Momentum conservation is described by the pressure drop during flowing. In trial calculation, when the flow rate rises from 0.3 mol/s to 0.6 mol/s, the *Re* number in the reaction zone ranges from 15,000 to 31,000, so the pressure drop in the reaction zone can be described by the Hicks equation [29,30,31]:(8)dpdz=−6.8(1−εp)1.2εp3ReR−0.2ρRcR2Dp
where *Re*_R_ is the *Re* number of the gas flow in the reaction zone; *c*_R_ is the flow rate, in m/s; *ρ*_R_ is the average density of the gas in the reaction zone, in kg/m^3^.

The pressure drop in the permeation zone is calculated by Darcy’s formula [32]:(9)dpPdz=64RePcP2ρP4R4

#### 2.3.3. Energy Conservation Equation

Local temperature change can be used to describe energy conservation. In the reaction zone:(10)$$\frac{dT}{dz}=\frac{P_{\mathrm{tube}}-H_{R}-H_{E}-H_{T}-H_{H}}{\sum_{k}F_{k}C_{p,k}}$$
where *H*_R_ is the heat absorbed by the ammonia decomposition reaction; *H*_E_ is the heat loss rate caused by the thermal radiation of the outer wall; *H*_T_ is the heat transferred from the reaction zone to the permeation zone; *H*_H_ is the energy loss caused by the permeation of hydrogen.
(11)$$H_{R}=S_{\mathrm{RA}}\left(1-\varepsilon_{p}\right)R_{A}\Delta_{r}H_{h}$$
(12)$$H_{E}=\rho_{s}\sigma T_{a}^{4}$$
(13)$$H_{T}=\frac{T-T_{P}}{HR_{\mathrm{RP}}}$$
(14)$$H_{H}=\int_{T_{P}}^{T}C_{p,H}dT$$

Since the energy in the permeation zone comes from reaction zone, the temperature change in the permeation zone is:(15)$$\frac{dT_{P}}{dz}=\frac{H_{T}+H_{H}}{\sum_{k}F_{k,P}C_{p,k}}$$

### 2.4. Performance Indicators

The conversion rate is an important performance indicator of the reactor which can measure the progress of the reaction. The ammonia conversion rate (ACR) is defined as:(16)$$ACR=1-\frac{F_{A,\mathrm{out}}}{F_{A,\mathrm{in}}}$$
where *F*_A,out_ and *F*_A,in_ are the outlet and inlet ammonia flow rates of the reaction zone, in mol/s.

If the reactor is used for solar hydrogen production, the hydrogen production rate (HPR) will be its core performance indicator:(17)$$HPR=F_{H,\mathrm{out}}+F_{H,P,\mathrm{out}}$$
where *F*_H,out_ and *F*_H,P,out_ are the outlet hydrogen molar flow rate in both the reaction and permeation zones, in mol/s.

The entropy generation rate (EGR) is an important performance indicator measuring the reversibility of a system or process:(18)$$EGR=S_{R}+S_{H,\mathrm{TR}}+S_{H,\mathrm{RP}}+S_{F,\mathrm{RA}}+S_{F,\mathrm{PA}}$$
where *S*_R_ is the entropy generation rate caused by chemical reaction; *S*_H,TR_ is the entropy generation rate in the process of heat transfer from the outer wall to the reaction gas; *S*_H,RP_ is the entropy generation rate of heat transfer from the reaction zone to the permeation zone; *S*_F_ is the entropy generation due to the flow pressure drop in the reaction zone; *S*_F,PA_ is the flow entropy generation in the permeate zone.

The local entropy generation rate of each process is [33,34,35,36,37]:(19)$$\sigma_{R}=S_{\mathrm{RA}}(1-\varepsilon p)R_{A}\frac{\Delta G}{T}$$
(20)$$\sigma_{H,\mathrm{TR}}=2\pi R_{2}q_{\mathrm{TR}}(\frac{1}{T}-\frac{1}{T_{a}})$$
(21)$$\sigma_{H,\mathrm{RP}}=2\pi R_{3}q_{\mathrm{RP}}(\frac{1}{T_{P}}-\frac{1}{T})$$
(22)$$\sigma_{F,\mathrm{RA}}=S_{\mathrm{RA}}c_{\mathrm{RA}}\frac{1}{T}\frac{dp}{dz}$$
(23)$$\sigma_{F,\mathrm{PA}}=S_{\mathrm{PA}}c_{\mathrm{PA}}\frac{1}{T_{P}}\frac{dp_{P}}{dz}$$
where *S*_RA_ and *S*_PA_ are the areas of the reaction and permeation zones, in m^2^; *c*_RA_ and *c*_PA_ are the gas flow velocities in the reaction and permeation zones, in m/s; *R*_A_ is the reaction rate of ammonia decomposition; Δ*G* is the Gibbs free energy of the reaction; *q_TR_* and *q_RP_* are the heat fluxes transferred from the reactor wall to the reaction gas and from the reaction zone to the permeation zone, respectively.

Therefore, the entropy generation rate of each process can be obtained by integrating the above local entropy generation rate over the length of the reactor.

Used in the field of solar energy storage, the heat absorption rate (HAR) is the core performance index of the heat storage reactor. It is the total amount of solar energy absorbed by the reactor:(24)$$HAR=L\cdot P_{\mathrm{tube}}-\int_{0}^{L}H_{E}dL$$
where *P*_tube_ is the power density of absorbed solar energy; *H*_E_ is the heat loss due to radiation.

As a chemical energy storage system, the energy conversion rate (ECR) of the endothermic reactor is also an important core indicator, which means that the energy converted into chemical energy accounts for a proportion of the total solar energy received:(25)$$ECR=\frac{\int_{0}^{L}H_{R}dL}{P_{\mathrm{sun}}K_{s}L}$$

## 3. Numerical Example

### 3.1. Performance Simulation of the Reference Reactor

According to the “Thermal Design Regulations for Civil Buildings” (JGJ24-86), the average ground solar radiation intensity is about 700 W/m^2^. So, when the light intensity is 700 W/m^2^, under the parameters in Table 2, the local flow rate of each component (Figure 3) and the temperature distribution (Figure 4) along the length of the reactor are obtained.

In Figure 3, ammonia decomposition reaction begins to occur within 1 m of the inlet section of the reactor. Because of the need to accumulate the partial pressure, obvious hydrogen permeation occurs only after 2 m. At the end of the reactor, ammonia is nearly completely decomposed, and the reaction rate is significantly reduced. At the same time, due to the existence of the permeation phenomenon, the partial pressure of hydrogen in the reaction zone began to decrease.

Corresponding to Figure 3, in Figure 4, since the ammonia decomposition reaction is not active in the first 1 m, the absorbed solar energy is mainly used to increase the temperature of the whole system. In the middle part of the reactor, the overall temperature tends to be stable because the ammonia decomposition reaction absorbs the solar energy. The temperature difference between the reaction zone and the permeation zone is small. However, at the end of the reactor, as the ammonia is nearly completely decomposed, the reaction rate drops and the temperature rise occurs again.

For the reference reactor, the ACR is 98.4%, the HPR is 0.74 mol/s, the EGR is 48.61 W/K, the HAR is 32.2 kW and the ECR is 61.6%.

### 3.2. Effect of Light Intensity on Reactor Performance

Solar energy is the energy source for the ammonia decomposition reaction, and the level of energy supply is a key external factor that directly determines the performance of the reactor. Therefore, under the premise of keeping other parameters as reference values, the changes in the performance indicators of the reactor as the surface light intensity increases from 500 W/m^2^ to 800 W/m^2^ are analyzed. Figure 5 and Figure 6 show the changes of the five performance indicators of ACR, HPR, EGR, HAR and ECR.

When the light intensity is 800 W/m^2^, the maximum HAR is 36.45 kW. From the numerical change in performance indicators, with the increase in light intensity, ACR, EGR, HAR and HPR increase by 36.2%, 39.6%, 58.5% and 36.2%, respectively, while ECR decreases by 14.2%.

From the perspective of the changing laws of the curves, in Figure 5 and Figure 6, except for the continuous growth of HAR, the curves of the remaining four indicators show an inflection point when the light intensity is about 650 W/m^2^. This is because, under the premise that the ammonia gas flow rate remains unchanged, the ammonia gas is almost completely decomposed when the light intensity reaches above 650 W/m^2^. Therefore, when the light intensity is above 650 W/m^2^, ACR and HPR remain basically unchanged, and ECR begins to decrease.

In Figure 7, when the light intensity is 650 W/m^2^, ammonia happens to be completely decomposed. So, the reactor temperatures keep stable after 2 m. When the light intensity is 800 W/m^2^, the reactor temperatures start to rise rapidly after 8 m, because, in this case, ammonia decomposes completely at 8 m, and after that, it can only rely on the increase in temperature to absorb solar heat.

When the light intensity is above 650 W/m^2^, the ammonia decomposition reaction has completed. So, all the solar energy needs to be absorbed by the temperature increase, which leads to an increase in heat transfer entropy generation. The entropy generation of the chemical reaction is close to zero at this time, so the *EGR* curve still rises, but the rising speed is much slower.

### 3.3. Effect of the Ammonia Flow Rate on Reactor Performance

As the working medium for energy storage or hydrogen production, the flow rate of ammonia is the most important operating parameter of the reactor and also has a decisive influence on the performance of the reactor. On the premise that the light intensity is 700 W/m^2^ and other parameters are taken as reference values, the changes of each performance index as the ammonia flow rate increases from 0.3 mol/s to 0.6 mol/s are analyzed. Figure 8 and Figure 9 show the variation of five performance indicators with the increasing ammonia flow rate.

When the ammonia flow rate is 0.6 mol/s, the maximum HAR is 32.39 kW. As the ammonia flow rate increases from 0.3 mol/s to 0.6 mol/s, the ECR, EGR, HAR and HPR increase by 73.6%, 71.7%, 15.6% and 74.3%, respectively, while ACR decreases by 12.9% during this process.

In Figure 8, ammonia is completely decomposed when the flow rate is below 0.45 mol/s, while the ACR began to decrease gradually with the increase in the ammonia flow rate. In Figure 8 and Figure 9, the curves of HPR, EGR and ECR all show inflection points when the ammonia flow rate is about 0.53 mol/s, rapidly changing from a linear increase to almost constant. This shows that the upper limit of HPR that can be provided by the light intensity of 700 W/m^2^ is about 0.78 mol/s. ECR is also stable when HPR is stable. EGR still has a slight upward trend after HPR stabilizes, because larger flow rates lead to higher flow entropy generation. In Figure 9, HAR gradually increases with the increase in the ammonia flow rate, but the growth rate becomes slower and slower.

### 3.4. Effect of the Membrane Radius on Reactor Performance

The radius of the outer wall of the reactor is determined according to the diameter of the heat absorption pipe of the LS-3 type heat collector, but there is no relevant design reference for the radius of the permeable membrane of the inner pipe. Therefore, under the condition that other parameters take reference values, Figure 10 and Figure 11 show the changes in performance indicators as the radius of the permeable membrane increases from 1 cm to 2 cm.

When the membrane radius is 1 cm, the maximum HAR is 32.22 kW. When the permeable membrane radius is increased from 1 cm to 2 cm, ACR, ECR, EGR and HPR increase by 1.3%, 1.4%, 0.27% and 1.3%, respectively, while HAR decreases by 0.1%. Except for EGR, all other indicators change slowly with the increase in the radius. The EGR increases slower and slower in the first part and then increases rapidly when the radius exceeds 1.7 cm. In the first half of the curve, EGR increases slowly with increasing ammonia conversion. However, once the radius exceeds 1.7 cm, the area of the reaction zone will be too low, resulting in a significant increase in flow entropy generation. When the radius increases from 1.5 cm to 1.7 cm, the flow entropy generation increases by 26.9%, while, when the radius increases from 1.7 cm to 1.9 cm, it increases by 42.4%.

### 3.5. Effect of the Nitrogen (Sweeping Gas) Flow Rate on Reactor Performance

In this model, the permeated hydrogen is removed by nitrogen co-current sweeping in the permeation zone. Therefore, the nitrogen flow rate will affect the hydrogen permeation rate, which in turn affects the reactor performance. The performance index changes that occur when the nitrogen flow rate is increased from 0.05 mol/s to 0.1 mol/s are shown in Figure 12 and Figure 13.

When the flow rate of nitrogen is 0.1 mol/s, the maximum HAR is 32.22 kW. The influence of nitrogen flow rate on the performance of the reactor is very weak. By increasing the nitrogen flow rate from 0.05 mol/s to 0.1 mol/s, the ACR and HPR decrease by about 0.1%. The higher nitrogen flow rate will take away more energy, which will cause the reaction temperature to become lower. For example, when the nitrogen flow rate increases from 0.05 mol/s to 0.07 mol/s to 0.09 mol/s, the outlet temperature of the reaction zone decreases from 701.5 K to 697.5 K to 693.4 K.

## 4. Conclusions

In this paper, the model of a solar-heated ammonia decomposition membrane reactor using nitrogen sweeping is established, and five performance indicators (ACR, HPR, EGR, HAR and ECR) are set considering the scenarios of clean energy hydrogen production and solar energy storage. The effects of the light intensity, ammonia flow rate, nitrogen flow rate and permeable membrane radius on system performance were analyzed according to these indicators. The conclusion is as follows:

For the reference reactor, the ACR is 98.4%, the HPR is 0.74 mol/s, the EGR is 48.61 W/K, the HAR is 32.2 kW and the ECR is 61.6%.

The two parameters of light intensity and ammonia flow rate have a very significant impact on the system performance, which can reach more than 70%. These two parameters will be the key optimization parameters in industrial design. The membrane radius has only a little effect on the system performance, with a performance change of about 1.2%. The effect of the nitrogen flow rate on system performance is less than 0.2%.

When the light intensity is increased from 500 W/m^2^ to 800 W/m^2^ (60% increase), the ACR, EGR, HAR and HPR increase by 36.2%, 39.6%, 58.5% and 36.2%, respectively, while the ECR decreases by 14.2%.

When the ammonia flow rate is increased from 0.3 mol/s to 0.6 mol/s (100% increase), the ECR, EGR, HAR and HPR increase by 73.6%, 71.7%, 15.6% and 74.3%, respectively, while the ACR decreases by 12.9%.

To avoid the premature complete decomposition of ammonia, the setting of the ammonia gas flow rate needs to match the light intensity.

## Figures and Tables

**Figure 1 membranes-12-00972-f001:**
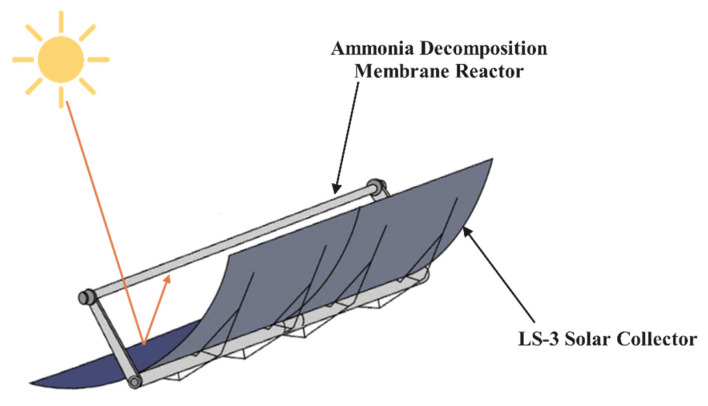
Model schematic of the ammonia decomposition membrane reactor.

**Figure 2 membranes-12-00972-f002:**
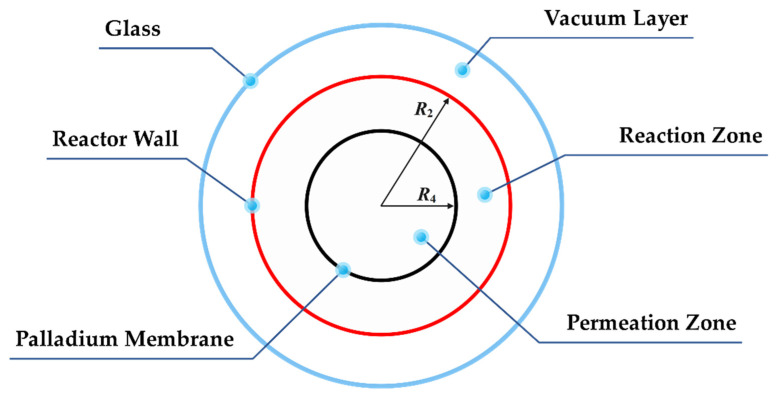
Cutaway view of the ammonia decomposition membrane reactor. (*R*_2_ is the inner radius of reactor outer wall; *R*_4_ is the inner radius of membrane).

**Figure 3 membranes-12-00972-f003:**
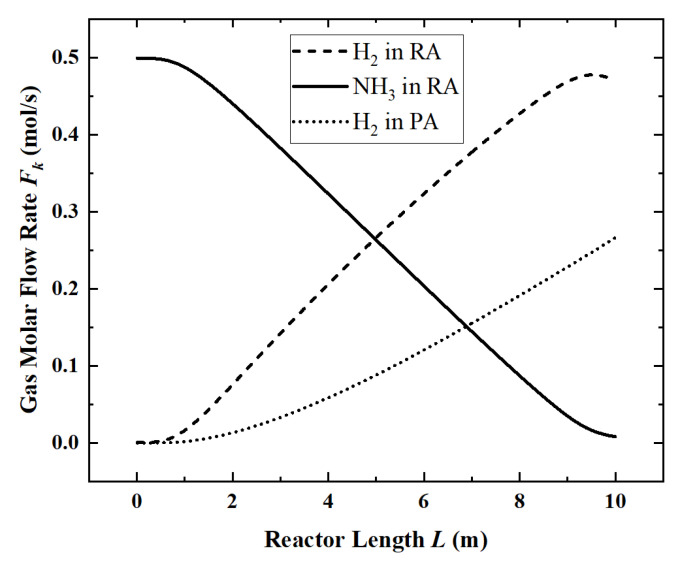
Molar flow rate of each component along the reactor length.

**Figure 4 membranes-12-00972-f004:**
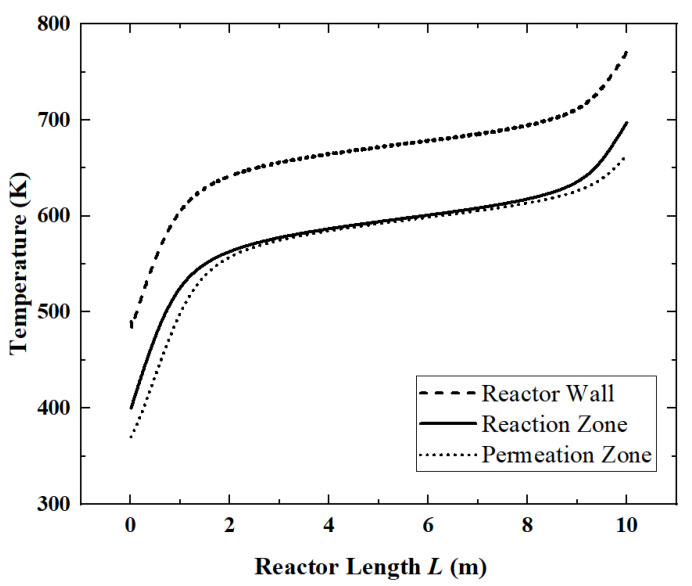
Temperature changes along the reactor length.

**Figure 5 membranes-12-00972-f005:**
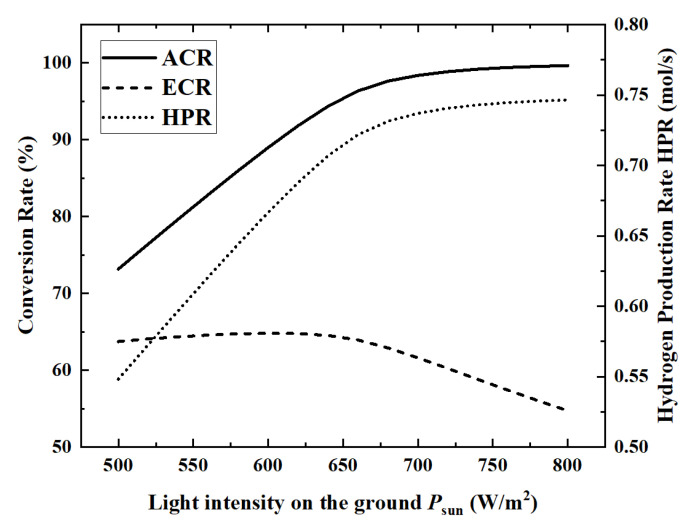
Effect of light intensity on ACR, ECR and HPR.

**Figure 6 membranes-12-00972-f006:**
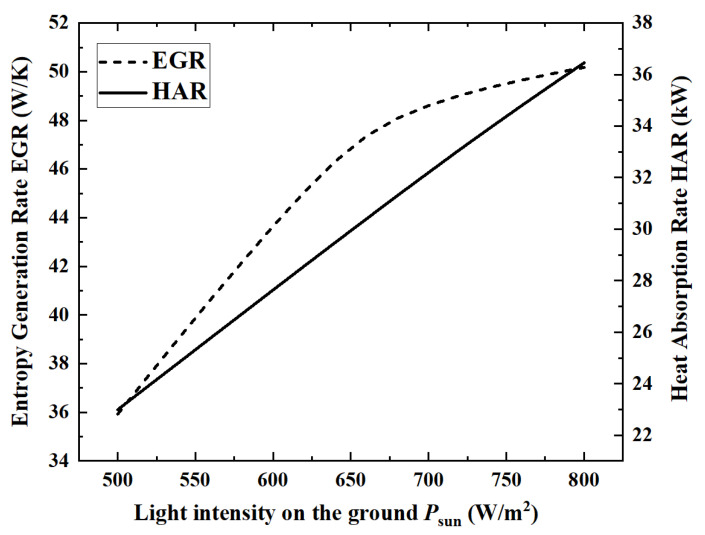
Effect of light intensity on EGR and HAR.

**Figure 7 membranes-12-00972-f007:**
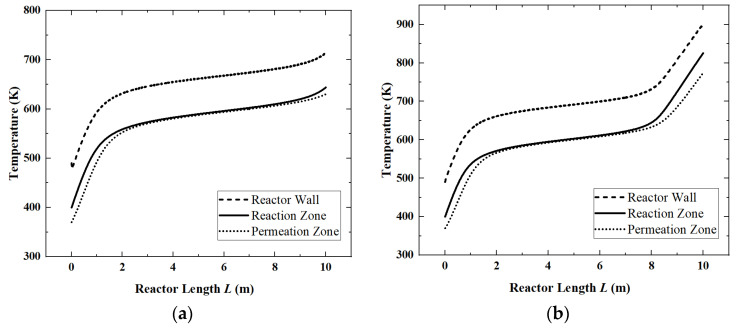
The temperature change of the reactor under different light intensities. (**a**) 650 W/m^2^; (**b**) 800 W/m^2^.

**Figure 8 membranes-12-00972-f008:**
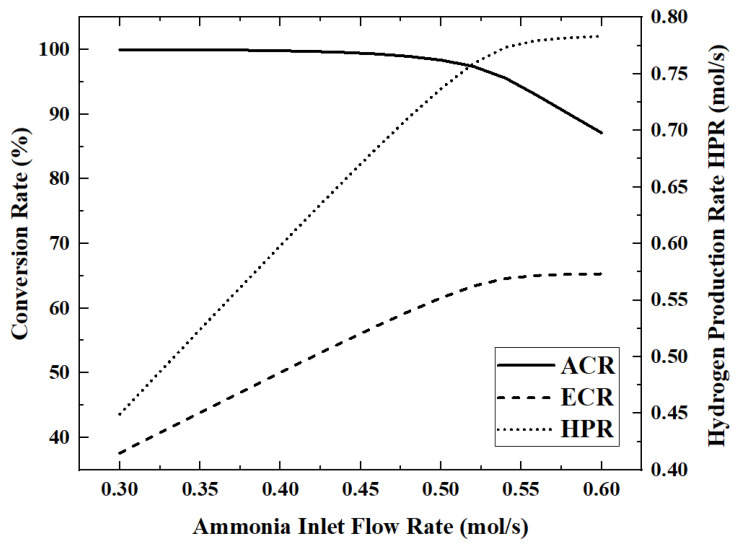
Effect of ammonia flow rate on ACR, ECR and HPR.

**Figure 9 membranes-12-00972-f009:**
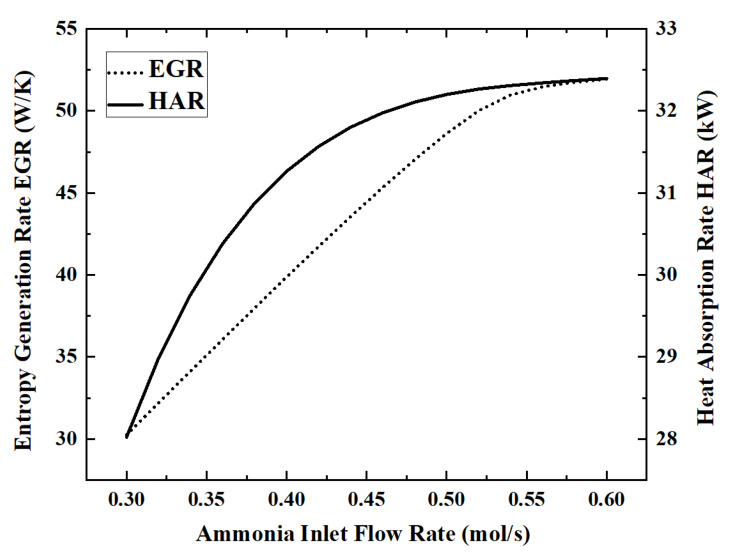
Effect of ammonia flow rate on EGR and HAR.

**Figure 10 membranes-12-00972-f010:**
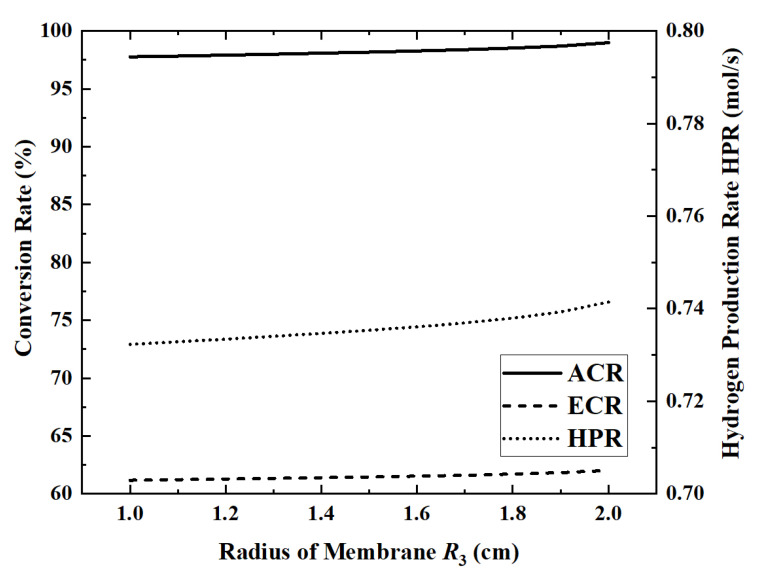
Effect of membrane radius on ACR, ECR and HPR.

**Figure 11 membranes-12-00972-f011:**
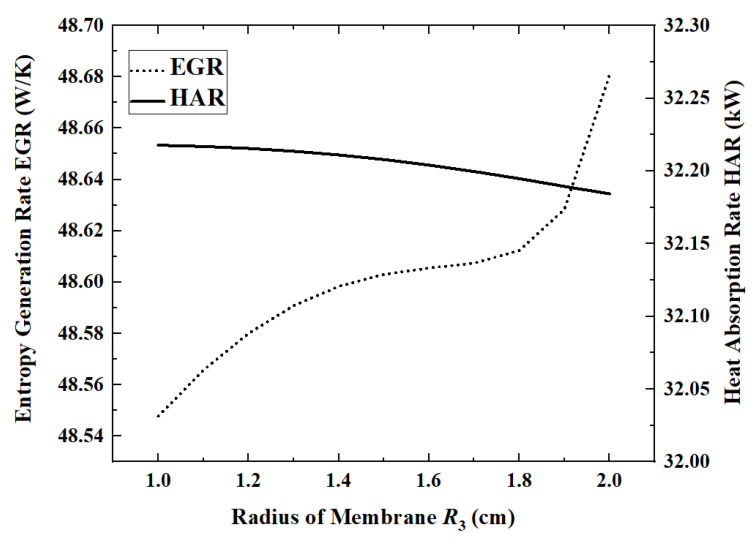
Effect of membrane radius on EGR and HAR.

**Figure 12 membranes-12-00972-f012:**
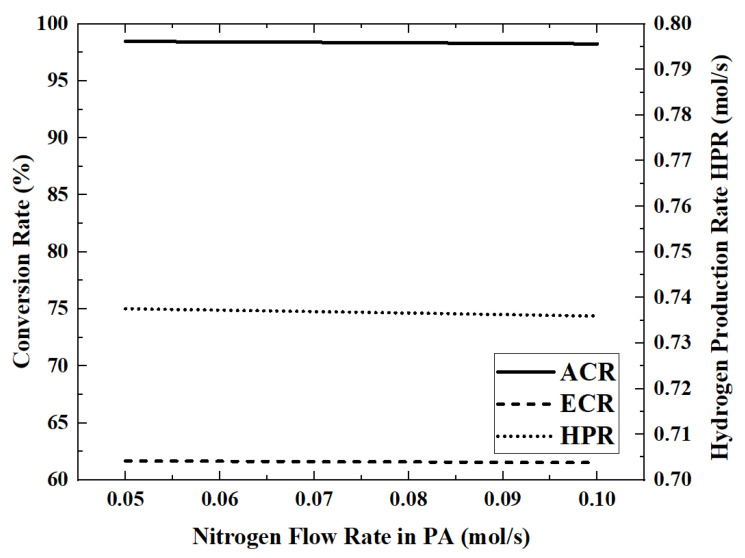
Effect of nitrogen flow rate on ACR, ECR and HPR.

**Figure 13 membranes-12-00972-f013:**
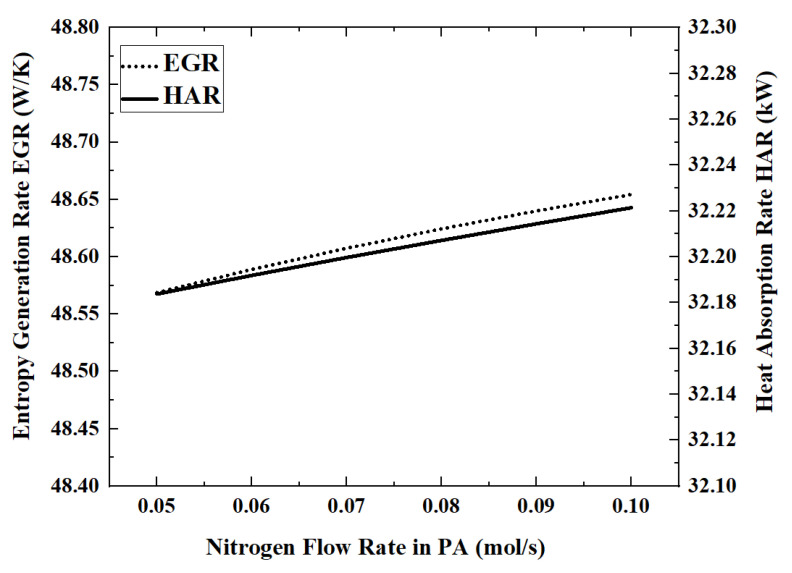
Effect of nitrogen flow rate on EGR and HAR.

**Table 1 membranes-12-00972-t001:** Parameters of the LS-3 solar collector.

Physical Parameters.	Symbols	Values
Width of collector	*K* _s_	5.76 m
Reflectivity of mirror	*η* _s_	90%
Light transmittance of glass outer tube	*δ* _s_	96%
Absorption rate of collector surface	*ε* _s_	96%
Collector surface emissivity	*ρ* _s_	10%

*K*_s_ is the opening width of the collector; usually, the wider the opening, the stronger the light-collecting ability. Sunlight entering the collector is focused by a parabolic mirror to its geometric focal point. There is a loss of energy in this process because the mirror reflectivity (*η*_s_) is not 100%. To reduce convective heat loss, there is a vacuum glass layer outside the reactor. Therefore, to pass through this glass layer, concentrated light will be affected by the light transmittance (*δ*_s_). The outer wall of the heat-absorbing tube of the concentrating heat collector has an optical coating, which can improve the absorption rate (*ε*_s_) of sunlight and reduce the emissivity (*ρ*_s_).

**Table 2 membranes-12-00972-t002:** Basic parameters of the reactor.

Physical Parameters	Symbols	Values
Reactor Length (m)	*L*	10
Outer radius of reactor outer wall (m)	*R* _1_	0.035
Inner radius of reactor outer wall (m)	*R* _2_	0.033
Outer radius of palladium membrane (m)	*R* _3_	0.017
Inner radius of palladium membrane (m)	*R* _4_	0.016
Membrane thickness (m)	*d_M_*	10^−5^
Ammonia inlet flow rate (mol/s)	*F* _Ain_	0.5
Nitrogen inlet flow rate (mol/s)	*F* _NPin_	0.07
Ammonia inlet pressure (bar)	*p* _in_	7
Nitrogen inlet pressure (bar)	*p* _Pin_	1

## Data Availability

Not applicable.

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
