# Peer review of "Performance Analysis of a Solar Heating Ammonia Decomposition Membrane Reactor under Co-Current Sweep"

_membranes, 2022, doi:10.3390/membranes12100972_

Round 1

Reviewer 1 Report

Title: Performance Analysis of Solar Heating Ammonia Decomposition Membrane Reactor under Co-current Sweep

Manuscript Nr.: Membranes-1919553

The authors present a simulation study, which focuses on the decomposition of ammonia either for H2 generation or heat storage by using a concentrating solar panel. The paper is in general easy to follow. The dependencies presented are in general quite known in the literature of palladium-based membrane reactors, however less in the context of a concentrating solar panel. I believe therefore the results to be of interest for the community. There are however several points to be tackled – namely, more thorough justification on the values use for the various parameters – before considering it for publication.

English / scientific language / notation

A non-exhaustive list is given below:

·         Line 36: one should read “by a butterfly solar concentrator”

·         Line 81: one should read “gas temperature in the reaction zone”

·         Line 138: one should read “in trial calculations, the Re number in the reaction ranges between 8000-30000”

·         Line 191: “absorbs a lot of heat” should be avoided.

·         In several instances the authors use “severe” without being clear why and what do they mean with it

·         Two different symbols are used for the Reynolds number (lines 139-141)

·         The authors introduce acronyms for quantities like hydrogen production rate (HPR) and ammonia conversion rate (ACR), which typically adopt one Greek or Latin letters. I had to move up and down in the paper several to always remember myself what is what (also because there were more acronyms for less familiar quantities). It would be advisable to use more reader friendly symbols for the quantities

·         The authors use in Fig. 4 “react area, permeation area” and in the text (line 192) the authors mention “reaction, permeation zone”. Consistency would be desired.

Introduction

·         Line 29: the authors mention that “the ammonia decomposition reaction has a significant absorption effect”, it should be mentioned beforehand that it is an endothermic reaction

·         Line 59: it would be good to have a simple explanation on what the terms “exergy efficiency, energy conversion rate and first law efficiency” mean

Scientific comments and questions

·         Although it becomes clear what is meant by “width of collector, glass outer tube” in Table 1 later in section 2.3, the information in Table 1 lacks accompanying text and having a more complete Figure 1 would help.

·         Why do the authors choose beta = 0.27 for their results? The numbers for k0 and Eapp also lack justification. In the literature there might be discrepancies in those values and it should be clear to the reader why the authors adopted these ones.

·         Lines 117 to 119: what do the authors mean by “severe reverse osmosis”? The authors should give numbers to justify this claim. Usually, sweeping in counter-current leads to higher permeation rates, so the authors should detail more what is happening in this configuration when compared to co-current.

·         Figure 2 should be updated with L, Rx information given in Table 2 for easier understanding. Moreover, this figure (or another one should be added) should make clear what the indices in equations (16)-(17) mean.

·         If I understand correctly: the hydrogen production rate is the sum of the hydrogen that permeates (mixed with N2) and that goes out in the retentate (mixed with N2 and with little NH3)? How do the authors envisage this to be implemented in light of a real hydrogen production system? Both retentate and permeate streams would be mixed into another stage to separate N2 from H2?

·         Why do the authors choose 10 m long reactor? It is unusually long when one compares to the data available in the literature of membrane reactors. Is this the length of the LS-3 concentrating collector?

·         Although the authors do not explicitly mention the membrane thickness, one can estimate from Table 2 that it is equal to 17 – 16 = 1 mm. This is quite a thick membrane. Do the parameters chosen in equation (7) correspond to a palladium membrane with this thickness? Alternatively, the authors should explain why they selected the parameters (Ek, Eact and n) of equation (7). Moreover, in this equation, what is the value taken for the membrane temperature TM? Perhaps the one at the reaction zone? It should be clear

·         The authors choose to use 7 bar for NH3 in the feed. This is quite a high pressure, for which significant energy should be expected. Why do the authors choose this pressure? High conversions are to be expected as well at lower feed pressures, especially with the ruthenium-based catalyst. To illustrate it, the authors could indeed show an additional section on pressure dependency.

·         Line 138: “in trial calculation, the Re number in the reaction ranges of 8000-30000”. The authors need to give more details about this trial run. Does this range correspond to the flow variation used in Figs. 7 & 8?

·         Section 3.1: why do the authors use 700 W/m2? Is this a typical value for solar panel concentrators? Does this correspond to the intensity reaching the collector or the one reaching the glass or even the reactor wall?

·         Section 3.1, line 188: What does it mean “since the ammonia decomposition is not severe in the first 1 meter” and what are its implications?

·         I believe in all plots where the hydrogen production rate is given the wrong acronym is used.

·         Section 3.2: How does the light intensity influences the temperature distributions in the reactor? This would help to understand Fig. 5.

·         Section 3.4: It is not clear why “once the radius exceeds 1.7 cm, the area of the reaction zone will be too low”? I would think that increasing the radius, increases the area of the palladium membrane and also the volume of the reaction zone, and as a result the resident time decreases and the permeation area increase, thereby increasing both NH3 conversion and hydrogen production rate (which is seen in Fig. 9), neglecting radial diffusion times. The authors should explain better why this leads to a sudden increase in the flow entropy generation.

·         Section 3.5: I would expect that an increase in N2 flow rate would contribute to smaller partial pressures in the permeate side and therefore increase the H2 permeation rate increase. However, it looks like this flow rate influences the ammonia decomposition rate and as a result it also decreases the hydrogen production rate. But why does the N2 sweep gas flow has an influence on NH3 decomposition rate?

Reviewer 2 Report

This manuscript can be considered to publish on Membranes.

Author Response

none.

Reviewer 3 Report

This paper addresses the finite-time thermodynamics to establish the ammonia decomposition membrane reactor model with the outermost vacuum insulation layer, the middle annular reaction zone, and the innermost layer a nitrogen-sweeping permeation zone. This work can be published after major revision.

1.    The abstract is not clear. Please, identify your results.

2.    The authors need to extend the literature survey.

3. We cannot find the objective of this study at the end of the introduction part of this paper.

4.  Experimental section needs more explanation for the experimental setup.

5. How did the authors monitor the solar temperature?

6. what is the maximum temperature you expect from the solar collector?

7. What is your software did you use for numerical simulation?

8. Can the authors compare the solar NH3 decomposition with other systems?

9. what is the maximum energy consumption of your system? why we should consider the decomposition of ammonia by solar energy, Pls explain.

10. Thermal catalytic NH3 decomposition occurs at 450 deg C in the existence of Ru/Al2O3. How much energy will you add to proceed NH3 decomposition?

11. Energy efficiency should be determined and compared with other systems.

12. Pls define your system in detail and maximum Absorption heat in the daytime.

13.   Did you measure the concentration of the outlet gas samples? How did you control air leakage to the system?

14. Pls validate your simulation work. Pls, change the title of the numerical part, "numerical example!".

15. In the conclusion part: the authors should tell us the optimum conditions. pls, explain your results.

16. English must be improved.

Round 2

Reviewer 1 Report

The authors addressed all points, although more efforts could have been done in the description of the Figures.

I still don't understand why the (outer membrane radius) - (inner membrane radius) = 0.017 - 0.016 = 0.001 m yields a thickness of 10 μm. Are we talking about a supported membrane? If so, please do write.

Before acceptance, I would still recommend a proof-reading of the manuscript by a native English speaker.

Author Response

Reply to comments for Membranes-1919553 from reviewer 1

Thank you for acknowledging our work.

And sorry for not describing so clearly on the membrane thickness.

Yes, you are right. The supported membrane is used in our case. Membrane reactors often use support structures, especially for such a long reactor in our article. Because the support structures are not included in our analysis, relevant description was missing.

And the description of supported membranes is added in the first paragraph in Section 3.1.

Wish you all the best!

Reviewer 3 Report

Congratulations to the authors. 

Author Response

Thank you for your constructive comments and affirmations on this article.